# Evidence of Exposure to SARS-CoV-2 in Dogs and Cats from Households and Animal Shelters in Korea

**DOI:** 10.3390/ani12202786

**Published:** 2022-10-15

**Authors:** Da-Yun Bae, Dongseob Tark, Sung-Hyun Moon, Jae-Ku Oem, Won-Il Kim, Chul Park, Ki-Jeong Na, Choi-Kyu Park, Yeonsu Oh, Ho-Seong Cho

**Affiliations:** 1College of Veterinary Medicine, Jeonbuk National University, Iksan 54596, Korea; 2Korea Zoonosis Research Institute, Jeonbuk National University, Iksan 54531, Korea; 3Special Committee on Veterinary Notifiable Infectious Diseases, Korean Veterinary Medical Association, Seongnam 13590, Korea; 4College of Veterinary Medicine, Chungbuk National University, Cheongju 28644, Korea; 5College of Veterinary Medicine, Kyungbuk National University, Daegu 41566, Korea; 6College of Veterinary Medicine & Institute of Veterinary Science, Kangwon National University, Chuncheon 24341, Korea

**Keywords:** COVID-19, SARS-CoV-2, cat, dog, zoonosis

## Abstract

**Simple Summary:**

With the worldwide spread of COVID-19, and given the zoonotic nature of this virus, it is necessary to monitor companion animals in epidemic areas. Dogs and cats visiting local animal hospitals and animal shelters during the COVID-19 pandemic were randomly selected for inclusion in this survey. Although testing is recommended only for animals from a COVID-19-confirmed family showing clinical symptoms, this investigation was carried out in view of the highly contagious nature of the virus. Of the 1018 pets tested, we found 2 cats with the SARS-CoV-2 antigen, 16 dogs (2.38%) and 18 cats (5.20%) carrying antibodies, and 14 dogs (2.08%) and 17 cats (4.91%) carrying virus-neutralizing antibodies. Even in animal shelters, which were thought to be relatively safe from the virus, with contact with confirmed human cases being blocked due to strict self-quarantine measures, antigens and antibody positivity were detected. Since antigens or antibodies were detected in the investigated animals regardless of whether the companion family was infected, their presence may cause continuous viral circulation between humans and animals, and the disease that started with zoonosis may cause reverse zoonosis. Considering the high transmissibility of COVID-19, continuous monitoring in animals is required.

**Abstract:**

The COVID-19 pandemic was caused by the zoonotic SARS-CoV-2. A variety of animals involved in human life worldwide have been investigated for infection. As the degree of infection increased, extensive monitoring in animals became necessary to determine the degree of infection in animals. The study was conducted on a sample of dogs and cats, which were randomly sampled according to the number of confirmed cases in the region. Animals from both COVID-19-confirmed households and generally disease-negative families and animal shelters were included. Tests included real-time qPCR tests for SARS-CoV-2 antigens, ELISA for antibodies, and plaque reduction neutralization tests (PRNT) for neutralizing antibodies. As a result, SARS-CoV-2 viral RNA was detected in 2 cats out of 1018 pets (672 dogs and 346 cats). A total of 16 dogs (2.38%) and 18 cats (5.20%) tested positive using ELISA, and 14 dogs (2.08%) and 17 cats (4.91%) tested positive using PRNT. Antigens of- and/or antibodies to SARS-CoV-2 were detected in the animals regardless of whether the companion family was infected; this was the case even in animal shelters, which have been regarded as relatively safe from transmission. In conclusion, continuous viral circulation between humans and animals is inevitable; therefore, continuous monitoring in animals is required.

## 1. Introduction

Coronavirus disease 2019 (COVID-19), caused by infection with SARS-CoV-2, is a human disease which most likely emerged from an animal source and through widespread human-to-human transmission became a pandemic. As of 31 July 2022, around 584 million confirmed human cases have been reported worldwide, with more than 6.4 million human deaths [1]. This new zoonotic virus has spread widely, indiscriminately leading to the infection of animals. Currently, 679 animal cases have been reported worldwide including 24 species, such as otters, pet ferrets, lions, tigers, pumas, snow leopards, and gorillas in 36 countries, after the first case of animal infection was reported in Hong Kong. Infected animals are known to present few clinical symptoms or pathological changes. Most of the reported infections in animals were originally transmitted from humans (i.e., reverse zoonosis) [2,3]. Recent reports of transmission from infected animals to humans warned of the possibility of reverse zoonosis. A veterinarian in Thailand was diagnosed with COVID-19 after sneezing while taking a swab sample from an infected cat owned by an infected patient [4]. A genetic study supported the hypothesis that SARS-CoV-2 was transmitted from the owner to the cat, and then from the cat to the veterinarian [4]. Reported transmissions from farm minks to farm workers in the Netherlands [5], and from a cat to a human in an animal hospital [4], indicate the possibility of other animal-to-human transmissions of SARS-CoV-2. In cases of animal shelters, the impact of COVID-19 is that more animals are being brought in and fewer are being adopted and rescued. Animal shelters, which must balance the public health of the community with the goal of saving animal lives, are experiencing staff shortages due to illness and self-quarantine [4].

Understanding the ecological and epidemiological roles of pets in the transmission of SARS-CoV-2 is critical for animal and human health; such development will aid in our identification of household reservoirs and our prediction of the potential enzootic maintenance of the virus. Many studies have suggested that dogs and cats are susceptible to severe acute respiratory syndrome coronavirus (SARS-CoV), while little is known about how susceptible they are to SARS-CoV-2 specifically, or its prevalence under natural conditions [1].

Although the infection status of dogs and cats has been studied in Italy, China, and the USA [6,7,8,9], the degree of infection in companion animals in many countries remains unknown. Research must be carried out to confirm the origin and natural reservoir of SARS-CoV-2 and to determine the role of other potential reservoirs and animal hosts. Moreover, investigation in this field is important for building a better understanding of the viral pathogenesis and host factors of the virus, as well as continuing to increase our knowledge and skills in order to obtain long-awaited vaccines and specific treatments [10].

This study was conducted to understand the degree of infection in companion animals in Korea during the pandemic era. As the degree of infection is widespread, and although testing is recommended only for animals within COVID-19-confirmed families showing clinical symptoms, it was thought that random sampling would be necessary to determine the degree of infection in animals in view of the highly contagious nature of the virus. Dogs and cats which visited local animal hospitals and animal shelters were randomly selected for the survey, according to the number of confirmed cases in the region. This study was conducted on animals from both COVID-19-confirmed households and generally households and disease-negative familiesand animal shelters during the COVID-19 pandemic.

## 2. Materials and Methods

### 2.1. Sample Collection

Dogs and cats which visited local animal hospitals and animal shelters were randomly selected for the survey. The demographic data of the animals were recorded, including breed, sex, age, recent medical history, the presence of respiratory signs (cough, sneezing, nasal discharge, and conjunctivitis), and whether they live with a COVID-19-positive owner. Nasopharyngeal/oropharyngeal swabs and blood samples were collected from a total of 1018 animals (672 dogs and 346 cats) in Korea between April 2021 and January 2022. The sample size was based on the human COVID-19-confirmed cases per 100,000 per province in Korea. As of March 2022, the human population of South Korea is 51,754,000, with 6,020,000 dogs and 2,580,000 cats [11]. No animal deaths due to SARS-CoV-2 were reported at the time of investigation. Swabs were immersed into a universal transport medium (UTM) (GDL Korea, Seoul, Korea). Blood samples were clotted and prepared as serum aliquots. All samples were transported immediately to the laboratory on ice and stored at −20 °C until processing. 

All samples were collected by personnel wearing full personal protective equipment, including head covers, goggles, N95 masks, gloves, and disposable gowns.

### 2.2. Virus and Cells

SARS-CoV-2 (IVCAS 6.7512) was isolated from a COVID-19 patient as previously described [12]. Vero E6 was purchased from ATCC (ATCC^®^CRL-1586™). All experiments of virus isolation and neutralizing (VN) testing using SARS-CoV-2 were performed under biosafety level 3 (BSL3) conditions.

### 2.3. Nucleic Acid Extraction and Reverse-Transcription Real-Time qPCR

All swab specimens were prepared for identifying SARS-CoV-2 RNA, according to the protocol below. The UTMs were sufficiently homogenized to make 200 μL aliquots for RNA extraction. The RNA was extracted using a 16TU-CV19 Viral DNA/RNA Prep Kit (MiCo BioMed, Seoul, Korea) and a Veri-Q PREP M16 instrument (MiCo BioMed, Korea) according to the manufacturer’s instructions. Briefly, a reverse-transcription real-time qPCR assay was conducted to detect the ORF3a and nucleocapsid (N) genes of SARS-CoV-2, using a commercial nCoV-QM PCR kit (MiCo BioMed) with a compatible instrument, Veri-Q PCR 316 QD-P100 (MiCo BioMed). For the PCR mixture, a 10 μL reaction contained 3 μL of master mix (polymerase, reverse transcriptase, buffer, and stabilizer), 1 μL of primer/probe mixture, 1 μL of internal positive control, and 5 μL of template RNA. The thermocycling conditions consisted of reverse transcription at 50 °C for 10 min, followed by initial denaturation at 95 °C for 3 min, followed by 45 cycles at 95 °C for 9 s, and at 58 °C for 30 s of denaturation, annealing, and extension, respectively. Cycle threshold (Ct) values under 40 were considered positive results for each gene (Table A1).

### 2.4. Enzyme-Linked Immunosorbent Assay (ELISA)

Antibody (Ab) tests were performed on the serum against the nucleocapsid protein (N protein) of the SARS-CoV-2 virus, using a commercial indirect ELISA (ID Screen^®^SARS-CoV-2 Double Antigen Multi-species, IDvet, Grabels, France). It was designed to add serum samples and N protein-recombinant antigen horseradish peroxidase (HRP) conjugate onto the microwell plates precoated with purified N protein-recombinant antigen. To determine the presence of SARS-CoV-2 Ab in the sera, the optical density (OD) value was measured at 450 nm. The assay was validated when the optical density of positive control (OD_PC_) was ≥0.35 and the mean of the positive control (OD_PC_) to negative control (OD_NC_) control ratio was greater than three. The optical density of each sample (OD_N_) was used to calculate the S/P ratio value (expressed as %) where *S*/*P* = 100*(OD_N_−OD_NC_)/(OD_PC_−OD_NC_). Samples tested using ELISA were considered positive if the *S/P* ratio was greater than 60% and doubtful when the *P/S* ratio ranged between 50 and 60%; meanwhile, samples displaying an *S/P* score lower than 50% in ELISA were considered negative [13].

The Canine Coronavirus (CCV) Antibody ELISA was prepared using a double-resistant one-step sandwich ELISA [9].

The ELISA was prepared using a double-resistant one-step sandwich ELISA [9]. The CCV antigen was pre-coated onto the microtiter plates and incubated overnight at 4 °C with shaking. Each serum at a 1:50 dilution was added and incubated for 90 min at 37 °C. Rabbit anti-dog IgG was added, and then freshly prepared substrates were used. TMB substrate was added to the plate for rendering color. The OD value at the wavelength of 450 nm was measured with an enzyme marker and was compared with a cut-off value (negative control value + 0.15) to determine the presence or absence of antibodies to CCV in the serum samples.

For Feline Coronavirus (FCoV) Antibody ELISA, microtiter plates were coated with 25 μg/mL of FCoV type II antigen (purified whole virus), and incubated overnight at 4 °C with shaking. Each serum at a 1:50 dilution was added and incubated for 90 min at 37 °C.

Microtiter plates were coated with 25 μg/mL of FCoV type II antigen (purified whole virus), and incubated overnight at 4 °C with shaking. Each serum at a 1:50 dilution was added and incubated for 90 min at 37 °C. Rabbit anti-cat IgG was added; then freshly prepared substrates were used. The ODs were determined at 450 nm. Negative sera (from uninfected SPF cats) were included to determine the ELISA cut-off values; sera with OD values higher than 5-fold the OD of negative sera were considered positive.

### 2.5. Plaque Reduction Neutralization Test (PRNT)

For measuring virus-neutralizing (VN) antibodies, the plaque reduction neutralization test (PRNT) is considered the gold standard for measuring neutralization antibodies among a couple of VN titration methods against SARS-CoV-2 [14,15]. The PRNT was carried out in the biosafety level 3 containment laboratory at the Korean Zoonosis Research Institute, Jeonbuk National University, Korea. Briefly, on the day, 0, 4 × 10^5^ Vero E6 cells/mL were seeded into 12-well plates and incubated for 20 h at 37 °C with 5% CO_2_. On day 2, virus thawing was performed followed by 10-fold serial dilutions from 10^−1^ to 10^−5^. Then, 2% low-temperature-melting agar was mixed using ultra-pure distilled water with a mixture of 2X minimal essential medium (MEM; Gibco, Life Technologies Corporation, Carlsbad, CA, USA) with 4% fetal bovine serum (FBS, Gibco, Life Technologies Corporation, Carlsbad, CA, USA) (1:1); this was washed twice with PBS. This was infected with 100μL of the diluted virus with triplication or more. Then, the mixture was incubated for 1 h at 37 °C with 5% CO_2_ with shaking at 10 min intervals. The infection media were eliminated. The mixture was washed with PBS; a final dispensing mixture with a final concentration of 1% low-temperature-melting agar, 1X MEM, and 2% FBS was obtained and incubated at room temperature for 10 min; then, it was incubated at 37 °C with 5% CO_2_ until the cytopathic effect (CPE) was observed. A measure of 1mL of 10% formalin was added to the overlayed agarose. This was fixed for 1.5 h after observing CPE. The staining material was removed. The virus presence was calculated with the following equation: Pfu/mL = no. of plaques/(Dilution × Infection volume(mL). According to the result, the dilution factor was chosen between 40 and 100 plaques [16].

### 2.6. Data Analysis

Summary statistics were calculated for the samples by region to assess the overall quality of the data, including normality. Fisher’s exact test [17] was used to analyze differences in antibody detection from households and animal shelters with known COVID-19 infection status, and antibody detection in dogs and cats. Spearman’s correlation [18] was used to analyze the relationship between human COVID-19 case numbers and the detection of antibodies in dogs and cats. All statistical analyses were performed using IBM SPSS Statistics (version 24, New York, NY, USA). Statistical significance was considered as *p* < 0.05.

## 3. Results

### 3.1. Detection of SARS-CoV-2 Antigen in Oropharyngeal and Nasal Samples of Dogs and Cats

The real-time qPCR amplification results of SARS-CoV-2 in samples include a positive control and a negative control (Figure A1). Amplification curves were produced from the N gene ([red line, Ct = 23.30]) and ORF3a gene ([blue line, Ct = 24.47]) of the SARS-CoV-2-positive control; the other amplification curves (yellow lines) display the IPC of the positive control (Ct = 22.16), the negative control (Ct = 21.43), and the negative sample (Ct = 21.34), respectively. The real-time qPCR results for SARS-CoV-2 are interpreted in Table A1.

Out of 1018 samples (672 dogs and 346 cats), 2 cats (household, Jeonbuk Province; household, Jeonnam Province) were found to be positive for SARS-CoV-2 RNA ([ORF3a, Ct = 24.241], [N gene = Ct = 24.316] and ([ORF3a, Ct = 29.853], [N gene = Ct = 29.472], respectively) from their swabs via real-time qPCR (Figure 1, Table 1 and Table 2).

### 3.2. Detection of SARS-CoV-2 Antibody in Serum Samples of Dogs and Cats

A total of 1018 samples (672 dogs and 346 cats) were tested for the detection of SARS-CoV-2 antibody with ELISA and PRNT. In ELISA results, a total of 34 samples from 10 provinces tested positive for antibodies against SARS-CoV-2 (Table 2). The samples from households were antibody-positive in 13 dogs and 16 cats. In the samples from animal shelters, three dogs and two cats were found to be positive for antibodies (Figure 1 and Table 1). The antibody-positive results showed no statistical significance among the regional factors or between household and animal shelter samples (*p* > 0.05). In addition, the antibody specificity was confirmed by performing a cross-reaction test between SARS-CoV-2-positive dog sera and inactivated CCV and between SARS-CoV-2-positive cat sera and inactivated FCoV using ELISA. The indirect ELISA showed good specificity since it had no serological cross-reactivity between the SARS-CoV-2 and CCV or between SARS-CoV-2 and FCoV.

In order to verify the neutralization efficiency of ELISA-positive sera, a PRNT assay for SARS-CoV-2 was performed. Of the serum-positive samples (n = 34), 31 samples showed neutralization activity with titers ranging from 1/24 to 1/45 (Table 2). Neutralization activity was not detected in three samples, which might be attributed to the lack of specific neutralizing epitopes.

Seropositivity among dogs and cats, split into risk factor groupings is presented in Table A2. There were no statistically significant differences among the risk factors (households, animal shelters, sex, and age). For provinces in Korea, there was no positive trend between the proportion of dogs that tested positive and the recorded burden of human disease (Spearman’s r = 0.261, *p* = 0.469), whereas there was a positive trend between the proportion of cats that tested positive and the recorded burden of human disease (Spearman’s r = 0.649, *p* = 0.049) (Figure 2).

## 4. Discussion

As with many other catastrophic pandemics, SARS-CoV-2, which was transmitted from an animal origin to humans, is causing devastating health and economic impacts worldwide [19]. To date, SARS-CoV-2 has been sporadically detected in naturally infected dogs and cats, most of which were living in close contact with infected humans [20,21,22]. Few studies of companion animals have been undertaken because of an inevitable research focus on human disease [23,24].

In Southern Italy, no SARS-CoV-2 viral RNAs were detected in companion animals in an investigation of 182 dogs and 313 cats. However, the VN to the virus was detected in 0.8% of dogs and 1.7% of cats tested [25]. Another study in the Netherlands detected 18.8% (17.3% of dogs and 20.4% of cats) cases of SARS-CoV-2 positivity (PCR- and/or antibody positive) out of 156 dogs and 152 cats living in households with at least one confirmed COVID-19-positive person, whereas the SARS-CoV-2 prevalence was much lower in 183 dogs and 140 cats that had simply visited a clinic (3.3% of dogs and 6.4% of cats) [26]. Meanwhile, in Poland, when the country was in the midst of the fourth wave of viral spread, companion animals showed relatively high seroprevalence (18.9% of the feline sera and 16.0% of the canine sera tested positive) [27]. The results suggested that animal cases are most likely related to the high case numbers in the human population; this indicates a continuous occurrence of trans-species virus transmissions from infected owners to their pets. The Ministry of Agriculture, Forestry, and Fisheries in Japan reported to the World Organisation for Animal Health (WOAH) that there were no clinical symptoms in dogs of COVID-19 patients who were positive until the fourth day of symptoms, and that the dogs tested negative when tested at the fifth day of owner’s symptoms. The entire genome of SARS-CoV-2 from infected minks was analyzed at a mink farm in the Netherlands, where clinical symptoms and mortality increased due to SARS-CoV-2 infection. The result was identified as a mutant virus, different from the reported one. The Dutch government reported to the WOAH that there was an infectious link between minks and humans; this was because the affected farmer’s family and the person who cared for them were also infected.

Our study found that seroconversion occurred in dogs and cats without a significant association to whether their human family members were infected. Even in animal shelters, which have been thought to be relatively safe from the virus because contact with confirmed human cases was blocked due to strict self-quarantine, antigens and antibody positivity were detected. It can be seen that animal shelters also pose a risk of SARS-CoV-2 infection. In relation to other cases of animal infections in a zoo [28], it was suggested that a human-animal interaction may promote interspecies transmission, because animal shelters were maintained by volunteers. Through current studies on SARS-CoV-2 animal infection, including this investigation, the transmission from infected people to animals, which can be termed reverse zoonosis, have been presented [6,9]. According to the data so far including the results of the present study the risk of animals spreading COVID-19 to people is considered to be low. In some situations, mostly during close contact, people have spread SARS-CoV-2 to certain types of animals, including pet dogs and cats. This human-animal virus sharing is sufficient to raise concerns about the emergence of new mutant strains.

In contrast to the serology results, all animals except one cat tested negative using a PCR, including animals living in households with confirmed COVID-19 human infection; these animals presented no typical respiratory symptoms in human COVID-19. These findings suggest that, although pet animals are seroconverted, viral shedding occurs only for a relatively short period of time. In experimental studies, cats stopped shedding the virus 10 days post-infection (dpi) and developed neutralizing antibody responses by 7–13 dpi [29,30]. Similar results were reported in an experimental infection of dogs, in which the virus was detected in feces up to 6 dpi, but not in oropharyngeal swabs [31]. However, in a naturally infected Pomeranian dog, SARS-CoV-2 RNA was detected from nasal swabs by quantitative RT-PCR for at least 13 days at low titers, whereas the virus was not detected in fecal/rectal samples [32], suggesting that virus-shedding patterns may vary in some animals. Half of the infected dogs had detectable antibodies by 14 dpi. Those studies and our results highlight similar challenges in detecting SARS-CoV-2 infection for both humans and animals [33]. 

To date, humans are the most potent source of SARS-CoV-2 transmission, not only to other humans, but also to animals. Based on our current knowledge, it is unlikely that infected pets play an active role in SARS-CoV-2 transmission to humans [4]. In contrast, there are several studies and communications on companion animals living in areas of high human infection [20,21]. Cats are susceptible to human SARS-CoV-2 infection likely due to the high degree of similarity between the human and feline forms of ACE2 [34]. Further investigation is needed regarding dog susceptibility to SARS-CoV-2. Additionally, many serological surveys in pets remain to be explored to reveal the extent of the transmission routes between COVID-19-infected humans and community pets [20,26,35].

The emergence of other zoonotic infections in the future is inevitable given the enormous diversity of pathogens, especially in wildlife, and ongoing viral evolution. Furthermore, the interaction between humans, animals, and the environment can promote their emergence, and result in further deadly pandemics [36,37].

Since antigens or antibodies were detected in the investigated animals regardless of whether the companion family was infected, this could cause continuous viral circulation between humans and animals; a disease that Started as zoonosismay lead to reverse zoonosis [4,19,25]. This is the first report of SARS-CoV-2 infection in companion animals in Korea. In light of the high transmission power of SARS-CoV-2, it is thought that disease monitoring in animals should be carried out in the future to predict infection patterns.

## 5. Conclusions

In conclusion, the present study found that pet animals such as dogs and cats can become infected with SARS-CoV-2; those animals may be reverse-infected by infected people. Although no country currently recommends routine SARS-CoV-2 testing in animals, the results of this study suggest that it is important to monitor dogs and cats for antibodies against SARS-CoV-2, in addition, the carrier or vector potential of companion animals should be studied in detail in order to see the actual role of these animals in the spread of SARS-CoV-2. Another pandemic situation may come in the future; therefore, an efficient surveillance system for animals living close to humans should be established.

## Figures and Tables

**Figure 1 animals-12-02786-f001:**
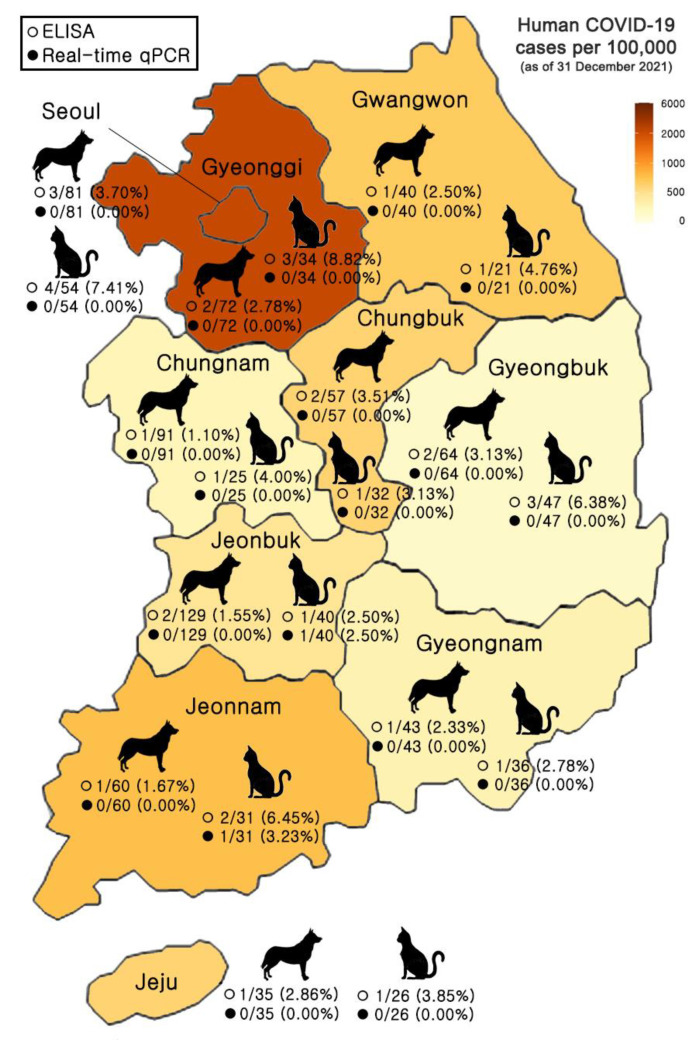
Distribution of dog and cat samples tested with real-time qPCR and ELISA across Korea. Data on human COVID-19 cases from the Ministry of Health and Welfare, 31 December 2021, and population data from the Ministry of the Interior and Safety, January 2020.

**Figure 2 animals-12-02786-f002:**
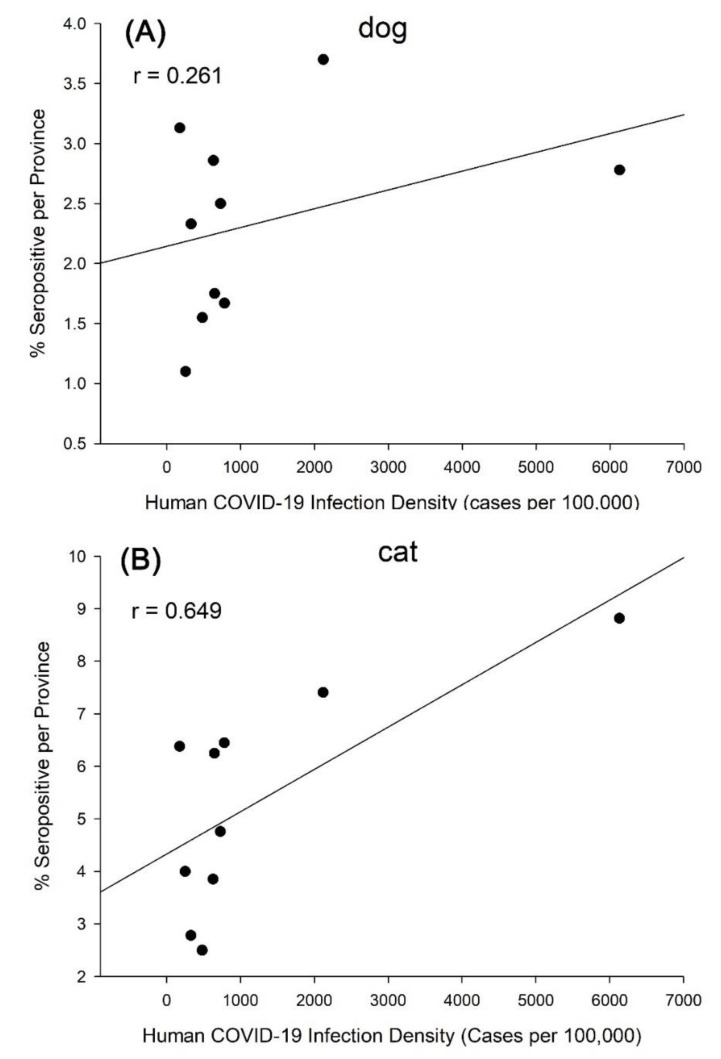
Correlation of percentage of seropositive dogs (**A**) and cats (**B**) per province and human COVID-19 infection in Korea.

**Table 1 animals-12-02786-t001:** Prevalence of SARS-CoV-2 in dogs and cats by regions in Korea.

Region (Province)	Type	qPCR		Total (%)	ELISA		Total (%)
		Dogs (%)	Cats (%)		Dogs (%)	Cats (%)	
Seoul	Households	0/81 (0.00)	0/54 (0.00)	0/135 (0.00)	3/81 (3.70)	4/54 (7.41)	7/135 (5.19)
	Animal Shelters	0	0	0	0	0	0
Gyeonggi	Households	0/58 (0.00)	0/31 (0.00)	0/89 (0.00)	1/58 (1.72)	3/31 (9.68)	4/89 (4.49)
	Animal Shelters	0/14 (0.00)	0/3 (0.00)	0/17 (0.00)	1/14 (7.14)	0/3 (0.00)	1/17 (5.88)
Gangwon	Households	0/40 (0.00)	0/21 (0.00)	0/61 (0.00)	1/40 (2.50)	1/21 (4.76)	2/61 (3.28)
	Animal Shelters	0	0	0	0	0	0
Chungbuk	Households	0/57 (0.00)	0/32 (0.00)	0/89 (0.00)	2/57 (3.51)	1/32 (3.13)	3/89 (3.37)
	Animal Shelters	0	0	0	0	0	0
Chungnam	Households	0/91 (0.00)	0/25 (0.00)	0/116 (0.00)	1/91 (1.10)	1/25 (4.00)	2/116 (1.72)
	Animal Shelters	0	0	0	0	0	0
Gyeonbuk	Households	0/40 (0.00)	0/37 (0.00)	0/77 (0.00)	1/40 (2.50)	1/37 (2.70)	2/77 (2.60)
	Animal Shelters	0/24 (0.00)	0/10 (0.00)	0/34 (0.00)	1/24 (4.17)	2/10 (20.00)	3/34 (8.82)
Gyeongnam	Households	0/43 (0.00)	0/36 (0.00)	0/79 (0.00)	1/43 (2.33)	1/36 (2.78)	2/79 (2.53)
	Animal Shelters	0	0	0	0	0	0
Jeonbuk	Households	0/56 (0.00)	1/35 (2.86)	1/91 (1.10)	1/56 (1.79)	1/35 (2.86)	2/91 (2.20)
	Animal Shelters	0/73 (0.00)	0/5 (0.00)	0/78 (0.00)	1/73 (1.37)	0/5 (0.00)	1/78 (1.28)
Jeonnam	Households	0/39 (0.00)	1/26 (6.25)	1/65 (1.82)	1/39 (2.56)	2/26 (7.69)	3/65 (4.62)
	Animal Shelters	0/21 (0.00)	0/5 (0.00)	0/26 (0.00)	0/21 (0.00)	0/5 (0.00)	0/26 (0.00)
Jeju	Households	0/35 (0.00)	0/26 (0.00)	0/61 (0.00)	1/35 (2.86)	1/26 (3.85)	2/61 (3.28)
	Animal Shelters	0	0	0	0	0	0
Subtotal	Households	0/540 (0.00)	2/323 (0.62)	2/863 (0.23)	13/540 (2.41)	16/323 (4.95)	29/863 (3.36)
	Animal Shelters	0/132 (0.00)	0/23 (0.00)	0/155 (0.00)	3/132 (2.27)	2/23 (8.70)	5/155 (3.23)
Total		0/672 (0.00)	2/346 (0.61)	2/1018 (0.20)	16/672 (2.38)	18/346 (5.20)	34/1018 (3.34)

**Table 2 animals-12-02786-t002:** Detection of antigens and antibodies against SARS-CoV-2 in dogs and cats by qPCR, ELISA, and PRNT.

No.	qPCR (Ct)		ELISA	PRNT	Background of Animal					
	ORF3a	N Gene	(OD450)	Neutralization Titer	Species	Sex	Age (year)	Province	Source	COVID-19 Patient Owner
1	>40	>40	0.7340	1/33	Cat	F	3	Seoul	Household	No
2	>40	>40	0.6376	1/24	Dog	M	8	Seoul	Household	No
3	>40	>40	0.9325	1/24	Cat	M	9	Seoul	Household	Yes
4	>40	>40	1.4351	1/45	Dog	M	5	Seoul	Household	No
5	>40	>40	1.2310	1/45	Cat	M	1	Seoul	Household	No
6	>40	>40	0.6832	1/33	Cat	F	4	Seoul	Household	No
7	>40	>40	0.7138	1/24	Dog	F	2	Seoul	Household	No
8	>40	>40	0.6241	1/33	Cat	M	1	Gyeonggi	Household	No
9	>40	>40	0.6968	1/24	Dog	F	3	Gyeonggi	Household	No
10	>40	>40	0.7120	1/33	Cat	F	3	Gyeonggi	Household	Yes
11	>40	>40	0.9221	1/45	Cat	M	4	Gyeonggi	Household	No
12	>40	>40	0.6793	1/24	Dog	M	Unknown	Gyeonggi	Animal shelter	No
13	>40	>40	0.7418	1/24	Dog	M	4	Gangwon	Household	No
14	>40	>40	0.6274	>1/5	Cat	F	2	Gangwon	Household	No
15	>40	>40	1.2123	1/33	Cat	F	3	Chungbuk	Household	No
16	>40	>40	1.8163	1/45	Dog	M	8	Chungbuk	Household	Yes
17	>40	>40	1.4465	1/33	Dog	M	9	Chungbuk	Household	No
18	>40	>40	0.6275	>1/5	Dog	M	5	Chungnam	Household	No
19	>40	>40	0.9278	1/33	Cat	M	4	Chungnam	Household	No
20	>40	>40	0.6431	1/24	Cat	M	1	Gyeongbuk	Household	Yes
21	>40	>40	0.8537	1/33	Dog	F	2	Gyeongbuk	Household	No
22	>40	>40	1.1062	1/33	Dog	M	Unknown	Gyeongbuk	Animal shelter	No
23	>40	>40	1.2013	1/33	Cat	F	1	Gyeongbuk	Animal shelter	No
24	>40	>40	1.0152	1/24	Cat	F	Unknown	Gyeongbuk	Animal shelter	No
25	>40	>40	0.9312	1/24	Dog	F	1	Gyeongnam	Household	No
26	>40	>40	1.1136	1/33	Cat	M	8	Gyeongnam	Household	No
27	>40	>40	0.8532	1/33	Dog	M	2	Jeonbuk	Household	Yes
28	24.241	24.316	1.5630	1/24	Cat	M	2	Jeonbuk	Household	Yes
29	>40	>40	0.7381	1/33	Dog	F	1	Jeonbuk	Animal shelter	No
30	29.853	29.472	1.5422	1/24	Cat	F	3	Jeonnam	Household	Yes
31	>40	>40	0.8922	>1/5	Dog	F	1	Jeonnam	Household	No
32	>40	>40	0.9273	1/33	Cat	M	3	Jeonnam	Household	No
33	>40	>40	1.4685	1/24	Dog	M	4	Jeju	Household	No
34	>40	>40	0.9014	1/33	Cat	F	2	Jeju	Household	No

## Data Availability

Not applicable.

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
