# Peer review of "Evidence of Exposure to SARS-CoV-2 in Dogs and Cats from Households and Animal Shelters in Korea"

_animals, 2022, doi:10.3390/ani12202786_

Round 1
Reviewer 1 Report
The uploaded article provides additional data about the SARS-CoV-2 exposure in samples obtained from dogs and cats in Korea.
Thanks to the authors for their work. The study is interesting but needs some changes to improve its presentation as an article.
- I recommend that authors check the italics and bold used in paragraph titles.
- I recommend that the authors check some typos at the lines 66, 301, 333.
- I suggest to remove the paragraphs "Patents" and "Supplementary materials " if unnecessary.
I suggest to remove the keywords prevalence and reverse zoonosis. I do not consider it the term prevalence properly correct to use considering that a calculation of the samples size needed to represent the canine and feline population has been made on human cases of COVID-19. Additionally, reverse zoonosis is not correct considering that no sequencing investigation has been done in this regard.
INTRODUCTION
- Line 60-62. Please provide the citation reference. In addition, i suggest rephrasing the sentence avoiding "sneezed on by an".
- Line 59-69 I suggest constructing better the sentences, so that it is made clearer the logical connection of the discourse carried on by the authors.
MATERIALS AND METHODS
- Line 141. considering the similarity of the text I recommend adding the reference:
Laidoudi Y, Sereme Y, Medkour H, Watier-Grillot S, Scandola P, Ginesta J, Andréo V, Labarde C, Comtet L, Pourquier P, Raoult D, Marié JL, Davoust B. SARS-CoV-2 antibodies seroprevalence in dogs from France using ELISA and an automated western blotting assay. One Health. 2021 Jul 18;13:100293. doi: 10.1016/j.onehlt.2021.100293. PMID: 34377760; PMCID: PMC8327341.
- Line 143 Please provide information about the ELISA kit.
- Line 145 Please provide time and temperature of incubation
- Line 154 Please provide the criteria of evaluation of results (as positive or negative sera) as previously realized for Sars Cov 2 Elisa and Canine CoV ELISA.
RESULTS
- Line 203 Please change the P of p value in lowercase and italic
- Figure 2 is indicated and commented on in the text?
DISCUSSION
- Line 262 Please provide citation references
- Line 264 Please provide citation references
- Line 267-301 I suggest reconstructing the paragraph by checking the sentences and making clearer the logical sense of the discourse the authors want to pursue.
Additionally, I also suggest that authors check out more recent work
(i.e. Kannekens-Jager et al. 2022 doi: 10.1111/tbed.14713.
Decaro et al 2022 doi: 10.1111/tbed.14308.
Murphy and Ly 2022 doi: 10.1002/jmv.28035.
Barroso-Arevalo wt al 2022 doi: 10.1111/tbed.14366.
Teixeira et al. 2022 doi: 10.1007/s15010-022-01860-5.
Bienzle et al 2022 doi: 10.3201/eid2806.220423.
Bessiere et al 2022 doi: 10.3390/v14061178.
Barroso et al 2022 doi: 10.3390/microorganisms10020345.
Cardillo et al 2022 doi: 10.3390/microorganisms10020263.)
- Line 314 why 'both humans and animals?What challenges are you referring to?
- Line 317 Please provide citation reference
- Line 318 Please provide citation reference
- Line 319 Please provide citation reference
- Line 323 Please provide citation reference
- Line 324-327 I think this part is a bit too strong in its conclusions and lacks bibliographic references
- Line 331 Please provide citation reference
Author Response
Response to reviewers’ comments
We are pleased to resubmit a revised manuscript (no. animals-1962072) entitled “Evidence of exposure to SARS-CoV-2 in dogs and cats from households and animal shelters in Korea” for reconsideration in Animals published by MDPI as an original manuscript. We have carefully evaluated the reviewer’s comments and have provided a point-by-point response below. Changes in the manuscript have been identified by colored font. We hope that the revised manuscript meets the reviewers’ expectations at Animals.
Comments and Suggestions for Authors #1.
The uploaded article provides additional data about the SARS-CoV-2 exposure in samples obtained from dogs and cats in Korea. Thanks to the authors for their work. The study is interesting but needs some changes to improve its presentation as an article.
- I recommend that authors check the italics and bold used in paragraph titles.
--> Thank you for the comment. All paragraph titles have been changed to italics and bold.
- I recommend that the authors check some typos at the lines 66, 301, 333.
--> Thank you for the comment. Some typos at the lines 66, 301, 333 were corrected.
- I suggest to remove the paragraphs "Patents" and "Supplementary materials " if unnecessary.
--> Thank you for the reviewer’s comment. The "Patents" and "Supplementary materials " have been removed according to the reviewer’s comment.
I suggest to remove the keywords prevalence and reverse zoonosis. I do not consider it the term prevalence properly correct to use considering that a calculation of the samples size needed to represent the canine and feline population has been made on human cases of COVID-19. Additionally, reverse zoonosis is not correct considering that no sequencing investigation has been done in this regard.
--> Thank you for the reviewer’s comment. The “Prevalence” and “Reverse zoonosis” in keywords have been removed according to the reviewer’s comment
INTRODUCTION
- Line 60-62. Please provide the citation reference. In addition, I suggest rephrasing the sentence avoiding "sneezed on by an".
--> Thank you for the comment. The reference is added in the text. The sentence has been revised according to the reviewer’s comment.
- A veterinarian in Thailand was diagnosed with COVID-19 after sneezing while taking a swab sample from an infected cat owned by an infected patient.
- Line 59-69 I suggest constructing better the sentences, so that it is made clearer the logical connection of the discourse carried on by the authors.
--> Thank you for the reviewer’s comment. The sentence has been revised according to the reviewer’s comment.
MATERIALS AND METHODS
- Line 141. considering the similarity of the text I recommend adding the reference:
Laidoudi Y, Sereme Y, Medkour H, Watier-Grillot S, Scandola P, Ginesta J, Andréo V, Labarde C, Comtet L, Pourquier P, Raoult D, Marié JL, Davoust B. SARS-CoV-2 antibodies seroprevalence in dogs from France using ELISA and an automated western blotting assay. One Health. 2021 Jul 18;13:100293. doi: 10.1016/j.onehlt.2021.100293. PMID: 34377760; PMCID: PMC8327341.
--> Thank you for the reviewer’s comment. The reference is added in the line 141.
- Line 143 Please provide information about the ELISA kit.
--> Information of CCV antibody ELISA was provided in the text.
- Line 145 Please provide time and temperature of incubation
--> Information of CCV antibody ELISA was provided in the text.
The ELISA was prepared by double-resistant one-step sandwich ELISA [9]. The CCV antigen was precoated onto the microtiter plates and incubated overnight at 4℃with shaking. Each serum in 1:50 dilution, was added and incubated for 90 min at 37℃. Rabbit anti-dog IgG was added, and then freshly prepared substrates were used.
- Line 154 Please provide the criteria of evaluation of results (as positive or negative sera) as previously realized for Sars Cov 2 Elisa and Canine CoV ELISA.
--> Negative sera (from uninfected SPF cats) were included to determine the ELISA cut-off values; sera with OD values higher than 5-fold the OD of negative sera were considered positive.
RESULTS
- Line 203 Please change the P of p value in lowercase and italic
--> The P of p value was changed in lowercase and italic
- Figure 2 is indicated and commented on in the text?
--> Figure 2 is indicated and commented on Line 221.
DISCUSSION
- Line 262 Please provide citation references
--> Thank you for the reviewer’s comment. The reference is added.
- Line 264 Please provide citation references
--> Thank you for the reviewer’s comment. The reference is added.
- Line 267-301 I suggest reconstructing the paragraph by checking the sentences and making clearer the logical sense of the discourse the authors want to pursue. Additionally, I also suggest that authors check out more recent work
--> Thank you for the reviewer’s comment. Recommended references (Kannekens-Jager et al. 2022 doi: 10.1111/tbed.14713. Decaro et al 2022 doi: 10.1111/tbed.14308. Murphy and Ly 2022 doi: 10.1002/jmv.28035. Barroso-Arevalo wt al 2022 doi: 10.1111/tbed.14366. Teixeira et al. 2022 doi: 10.1007/s15010-022-01860-5. Bienzle et al 2022 doi: 10.3201/eid2806.220423. Bessiere et al 2022 doi: 10.3390/v14061178. Barroso et al 2022 doi: 10.3390/microorganisms10020345. Cardillo et al 2022 doi: 10.3390/microorganisms10020263.) have been cited and the sentences have been re-written.
- Line 314 why 'both humans and animals? What challenges are you referring to?
--> Thank you for the reviewer’s comment. The sentence has been revised.
Based on these study results, it was explained that the SARS-CoV-2 infected animals shown in this study showed a relatively low antigen detection rate compared to the antibody-positive rate.
- Line 317 Please provide citation reference
--> Thank you for the reviewer’s comment. The reference is added.
- Line 318 Please provide citation reference
--> Thank you for the reviewer’s comment. The reference is added.
- Line 319 Please provide citation reference
--> Thank you for the reviewer’s comment. The reference is added.
- Line 323 Please provide citation reference
--> Thank you for the reviewer’s comment. The reference is added.
- Line 324-327 I think this part is a bit too strong in its conclusions and lacks bibliographic references
--> Thank you for the reviewer’s comment. The references are added.
- Carlson, C.J.; Albery, G.F.; Merow, C.; Trisos, C.H.; Zipfel, C.M.; Eskew, E.A.; Olival, K.J.; Ross, N.; Bansal, S. Climate change increases cross-species viral transmission risk. Nature. 2022, 607, 555-562. doi: 10.1038/s41586-022-04788-w.
- Huggel, C.; Bouwer, L.M.; Juhola, S.; Mechler, R.; Muccione, V.; Orlove, B.; Wallimann-Helmer, I. The existential risk space of climate change. Clim. Change. 2022, 174, 8. doi: 10.1007/s10584-022-03430-y.
- Line 331 Please provide citation reference.
--> Thank you for the reviewer’s comment. The reference is added.
Reviewer 2 Report
This article fills in the gap of testing and diagnostics of companion animals, which is very helpful for people to understand COVID-19 transmission in the animal population. The article is well-written and informative. I have the following suggestions for your consideration:
Line 96, how were the animals randomly selected? What was the statistical method?
Line 111, “All virus infection experiments…”, do you mean “virus isolation” instead?
Line 111, add the origin (city, country, etc) of the products/company.
Line 155, how does the plaque reduction neutralization test (PRNT) compare with Spearman-Karber or Reed-Muench method?
Line 177, consider adding a citation on the Fisher’s exact test for readers’ convenience. Same for Line 179, spearman’s correlation.
Line 346 to 347 may need to be removed as this is from journal instructions.
For Figure 1, some background information may be helpful for the readers to put this research into context. What are the estimated total numbers of domestic cats and dogs in Korea? What is the human population density and the companion animal population density?
Optional discussion points: Were there any reported deaths of animals due to COVID-19 infection? Were there any known animal-to-human transmission of SARS-CoV-2?
Author Response
Response to reviewers’ comments
We are pleased to resubmit a revised manuscript (no. animals-1962072) entitled “Evidence of exposure to SARS-CoV-2 in dogs and cats from households and animal shelters in Korea” for reconsideration in Animals published by MDPI as an original manuscript. We have carefully evaluated the reviewer’s comments and have provided a point-by-point response below. Changes in the manuscript have been identified by colored font. We hope that the revised manuscript meets the reviewers’ expectations at Animals.
Reviewer #2
This article fills in the gap of testing and diagnostics of companion animals, which is very helpful for people to understand COVID-19 transmission in the animal population. The article is well-written and informative. I have the following suggestions for your consideration:
Line 96, how were the animals randomly selected? What was the statistical method?
--> Our “randomly selected” means randomly selected with a probability of selection proportional to the number of animal hospitals and animal shelters per province. The detail was complemented in the sample collection of the Materials and Methods section.
Line 111, “All virus infection experiments…”, do you mean “virus isolation” instead?
--> The sentence was intended to make sure that experiments of virus isolation and neutralizing (VN) test using SARS-CoV-2 were performed under biosafety level 3 (BSL3) conditions.
Line 111, add the origin (city, country, etc) of the products/company.
--> Thank you for the reviewer’s comment. Information of the company was added.
Line 155, how does the plaque reduction neutralization test (PRNT) compare with Spearman-Karber or Reed-Muench method?
--> Liu et al. (2022) described the plaque reduction neutralization test (PRNT) is considered the gold standard for measuring neutralization antibodies against SARS-CoV-2 in human patient sera. We did not compare other methods.
Liu, K.-T.; Han, Y.-J.; Wu, G.-H.; Huang, K.-Y.A.; Huang, P.-N. Overview of Neutralization Assays and International Standard for Detecting SARS-CoV-2 Neutralizing Antibody. Viruses 2022, 14, 1560. https:// doi.org/10.3390/v14071560
Line 177, consider adding a citation on the Fisher’s exact test for readers’ convenience. Same for Line 179, spearman’s correlation.
--> References on the statistical methods used have been added according to the reviewer’s comment. Thank you.
Fisher’s exact test [17] was used to analyze differences in antibody detection from households and animal shelters with known COVID-19 infection status, and antibody detection from dogs and cats. Spearman’s correlation [18] was used to analyze the relationship between human COVID-19 case numbers and detection of antibodies in dogs and cats.
Line 346 to 347 may need to be removed as this is from journal instructions.
--> Thank you for the reviewer’s comment. The sentence has been removed.
For Figure 1, some background information may be helpful for the readers to put this research into context. What are the estimated total numbers of domestic cats and dogs in Korea? What is the human population density and the companion animal population density?
--> Thank you for the comment. The content you mentioned has been added in the Introduction section.
The sample size was based on the human COVID-19-confirmed cases per 100,000 per province in Korea. As of March 2022, the human population of South Korea is 51,754,000, with 6,020,000 dogs and 2,580,000 cats [11].
- e-Index. e-Indicators in South Korea. http://www.index.go.kr/main.do (Accessed 5 June 2021).
Optional discussion points: Were there any reported deaths of animals due to COVID-19 infection? Were there any known animal-to-human transmission of SARS-CoV-2?
--> As the reviewer commented, the “No animal deaths due to SARS-CoV-2 were reported at the time of investigation”. The sentence was included in the manuscript. Thank you.
Reviewer 3 Report
The manuscript entitled Evidence of exposure to SARS-CoV-2 in dogs and cats from households and animal shelters in Korea is very well written and explained. The introduction is written in a manner that attracts the attention of reader, furthermore it highlights the importance of the study. In the section of material and methods all planed work is explained in an easy manner with complete details. Results are very well explained. Data clearly elaborates the findings of study. In the discussion present findings are explained in details with the past findings in literature. However, I have one suggestion regarding Conclusion that the suggestion regarding the carrier/vector potential of pet/companion animals should be studied in detail in order to see the actual role of these animals in the spread of SARS-CoV-2. Best wishes.
Author Response
Response to reviewers’ comments
We are pleased to resubmit a revised manuscript (no. animals-1962072) entitled “Evidence of exposure to SARS-CoV-2 in dogs and cats from households and animal shelters in Korea” for reconsideration in Animals published by MDPI as an original manuscript. We have carefully evaluated the reviewer’s comments and have provided a point-by-point response below. Changes in the manuscript have been identified by colored font. We hope that the revised manuscript meets the reviewers’ expectations at Animals.
Reviewer #3
I have one suggestion regarding Conclusion that the suggestion regarding the carrier/vector potential of pet/companion animals should be studied in detail in order to see the actual role of these animals in the spread of SARS-CoV-2.
--> Thank you for the reviewer’s meticulous advice. The sentence has been revised according to the reviewer’s kind comment.